# Chest CT Super-resolution and Domain-adaptation using Memory-efficient 3D Reversible GANs

**Tycho F. A. van der Ouderaa**[1,2]                                          tychovdo@gmail.com

**Daniel E. Worrall**[1]                                                         d.e.worrall@uva.nl

**Bram van Ginneken**[2]                          bram.vanginneken@radboudumc.nl

[1] *University of Amsterdam, Amsterdam, The Netherlands*

[2] *Diagnostic Image Analysis Group, Radboud UMC, Nijmegen, The Netherlands*

## 1. Introduction

Recently, paired (e.g. Pix2pix (Isola et al., 2017)) and unpaired (e.g. CycleGAN (Zhu et al., 2017)) image-to-image translation methods have shown effective in medical imaging tasks (Yi et al., 2018) (Wolterink et al., 2017). In practice, however, it can be difficult to apply these deep models on medical data volumes, such as from MR and CT scans, since such data volumes tend to be 3-dimensional and of a high spatial resolution pushing the limits of the memory constraints of GPU hardware that is typically used to train these models.

Recent studies in the field of invertible neural networks have shown that reversible neural networks do not require to store intermediate activations for training (Gomez et al., 2017). We use the RevGAN model (van der Ouderaa and Worrall, 2019) that makes use of this property to perform memory-efficient partially-reversible image-to-image translation. We demonstrate this by performing a 3D super-resolution and a 3D domain-adaptation task on chest CT data volumes.

## 2. Problem Statement

The tasks addressed in this paper relates to CT imaging of the chest. Recent scanners allow for high-resolution scans with isotropic voxel sizes around or below 0.5 $mm^3$ and this requires the reconstruction of image matrices consisting of slices of 1024×1024 pixels, contrary to the traditionally used slices of 512×512 pixels. In these high-resolution scans, fine parenchymal details such as small airway walls, vasculature and lesion texture, are better visible. At the same time, the vast majority of scans available today for training networks consist of 512 matrices and often thicker slices, around 2mm. In addition, a wide variety of CT reconstruction kernels are used in practice, from more noisy high frequency kernels to smoother (soft kernels), and both traditional filtered backprojection as well as iterative reconstruction methods, varying between scanner vendors, are used (Mets et al., 2012). To produce robust and accurate image processing results, it is therefore desirable to pre-process chest CT scans to a standardized high resolution, and to remove the structural differences resulting from the use of different reconstruction algorithms. We aim to provide a proof-of-principle that we can use reversible networks to create a model capable of pre-processing CT scans to high resolution with a standardized appearance.

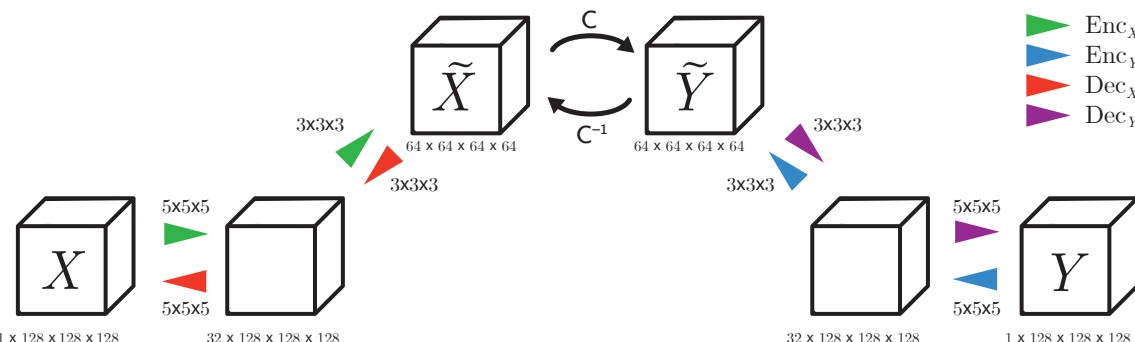

Figure 1: Illustration of the model: Encoders $\text{Enc}_X$ (green) and $\text{Enc}_Y$ (blue) lift/encode from the image space features spaces $\tilde{X}$ and $\tilde{Y}$. Decoders $\text{Dec}_X$ (red) and $\text{Dec}_Y$ (purple) project/decode back to $X$ and $Y$. The invertible mapping $C$ transforms between $\tilde{X}$ and $\tilde{Y}$.

## 3. Method

To perform memory-efficient image-to-image translation in 3D, we adapt the RevGAN model (van der Ouderaa and Worrall, 2019) that combines the Pix2pix loss (Isola et al., 2017) in Equation 1 for paired training or the CycleGAN loss (Zhu et al., 2017) in Equation 2 for unpaired training with a 3D architectures (Figure 1) and a reversible core to lower memory usage during training. For a more extensive description of the model losses, including a description of $L_{\text{GAN}}$, $L_{\text{L1}}$ and $L_{\text{cyc}}$, we refer to the original RevGAN paper.

**Paired Loss** For paired training, we combine the $L_{\text{GAN}}$ loss with the $L_1$ loss:

$$
\begin{aligned}
L_{\text{paired}}(F, G) = {} & L_{\text{GAN}}(G, D_X) + L_{\text{GAN}}(F, D_Y) \\
& + \lambda(L_{\text{L1}}(F, X, Y) + L_{\text{L1}}(G, X, Y))
\end{aligned}
\tag{1}
$$

**Unpaired Loss** For unpaired training, we combine the $L_{\text{GAN}}$ loss with the $L_{\text{cyc}}$ loss:

$$
\begin{aligned}
L_{\text{unpaired}}(F, G) = {} & L_{\text{GAN}}(G, D_X) + L_{\text{GAN}}(F, D_Y)) \\
& + \lambda L_{\text{cyc}}(G, F)
\end{aligned}
\tag{2}
$$

The $\lambda$ parameter determines the relative importance of the terms in the losses.

## 4. Data

**Super-resolution** The *Super-resolution* dataset contains 18 train volumes and 5 test volumes each consisting of 671 ($\pm 49$) slices of size $1024 \times 1024$ obtained with a high-end Canon CT scanner (Aquilion ONE). We train for 125 epochs on $128 \times 128 \times 128$ patches that were normalized in the range [-1, 1], uniformly redistributed from [-1150, 350] Hounsfield Units (HU). The source domain was generated by aggressively down-sampling 4 times in the z-dimension and 2 times in the x and y dimensions (corresponding to a typically used resolution of 512 by 512).

**Domain-adaptation** The *Domain-adaptation* dataset contains 17 CT scans for training and 3 CT scans for testing from The National Lung Screening Trial (NLST) (Team, 2011). All scans were obtained by a Siemens scanner and both a smooth (B30f) and a sharper (B50f) reconstruction kernel were available. Each scan contains an average of 169 ($\pm 14.7$) axial $512 \times 512$ slices. We train for 125 epochs on $64 \times 64 \times 64$ patches that were normalized in a range of [-1, 1], uniformly redistributed from [-1150, 350] HU.

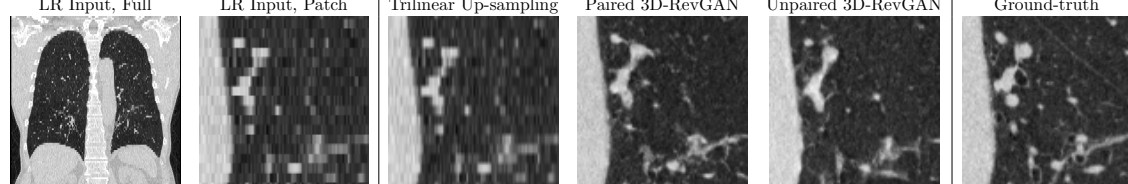

| LR Input, Full | LR Input, Patch | Trilinear Up-sampling | Paired 3D-RevGAN | Unpaired 3D-RevGAN | Ground-truth |

Figure 2: Visualization of a full and patched coronal slice of the first volume in the *Super-resolution* test set. From left to right: low-resolution input, trilinear up-sampling, paired 3D-RevGAN, unpaired 3D-RevGAN and high-resolution ground-truth.

## 5. Results

**Qualitative Results**   In Figure 2, we present a comparison between trilinear up-sampling, paired RevGAN and unpaired RevGAN. Upon closer inspection of the more detailed structures in the CT volumes, we find that the model produces visually compelling results and often is able to correctly fill-in parenchymal details such as small airway walls, vasculature and tissue texture. The output, however, is still far from perfect, which we account to the fact that the input has been down-sampled heavily. We expect even better results when the input source images are of a higher resolution.

**Quantitative Results**   As we can see from Table 1, we are able to increase performance by adding (+R) reversible blocks, without increasing memory cost. In Table 2, we show memory savings resulting from reversible layers.

| Model | Super-resolution (LR→HR) | | | Domain-adaptation (B50f→B30f) | | |
|---|---|---|---|---|---|---|
| | MAE | PSNR | SSIM | MAE | PSNR | SSIM |
| Unpaired | $0.24 \pm 0.007$ | $15.43 \pm 0.39$ | $0.44 \pm 0.008$ | $0.14 \pm 0.003$ | $14.56 \pm 0.22$ | $0.28 \pm 0.004$ |
| Unpaired+2R | $\mathbf{0.23} \pm 0.014$ | $\mathbf{16.43} \pm 0.44$ | $\mathbf{0.50} \pm 0.024$ | $\mathbf{0.13} \pm 0.003$ | $\mathbf{17.44} \pm 0.20$ | $\mathbf{0.29} \pm 0.003$ |
| Paired | $0.18 \pm 0.001$ | $15.89 \pm 0.16$ | $0.46 \pm 0.008$ | $0.14 \pm 0.001$ | $18.29 \pm 0.15$ | $0.33 \pm 0.007$ |
| Paired+2R | $\mathbf{0.15} \pm 0.000$ | $\mathbf{18.19} \pm 0.08$ | $\mathbf{0.48} \pm 0.008$ | $\mathbf{0.10} \pm 0.001$ | $\mathbf{23.24} \pm 0.16$ | $\mathbf{0.46} \pm 0.009$ |

Table 1: Mean and variance[2] of MAE, PSNR and SSIM scores. We can improve performance, using deeper models at the same level of memory complexity as shallow models.

| Depth | Model | **Activations (Ours)** | Activations (Naive) |
|---|---|---|---|
| 0 | 630.0 | **+ 2312.4** | (+ 2316.6) |
| 1 | 644.0 | **+ 2312.4** | (+ 2719.2) |
| 2 | 652.0 | **+ 2312.4** | (+ 3919.3) |
| 4 | 700.0 | **+ 2312.4** | (+ 4319.2) |
| 8 | 748.0 | **+ 2312.4** | (+ 5519.5) |

Table 2: Memory usage (MiB) of 3D-RevGAN model on GPU on a Nvidia Tesla K40m GPU. Note that the memory cost to store activations stays constant for deeper models.

## 6. Conclusion

This study provides a proof-of-principle for learned 3D pre-processing of CT scans to high resolution with a standardized appearance. We use the RevGAN model to successfully perform memory-intensive 3D domain-adaptation and 3D super-resolution tasks, using paired and unpaired data. Thereby, this study provides additional evidence that reversibility can be useful in practical settings to lower memory requirements of deep models. Future research should determine how well the model generalizes beyond the data used in this study.

## Acknowledgements

We are grateful to the Diagnostic Image Analysis Group (DIAG) of the Radboud University Medical Center and the Netherlands Organisation for Scientific Research (NWO) for supporting this research and providing computational resources.

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
