# OpenReview forum: "Chest CT Super-resolution and Domain-adaptation using Memory-efficient 3D Reversible GANs"
_MIDL.io/2019/Conference/Abstract — MIDL Abstract 2019_

### Official Review · AnonReviewer2 · 2019-04-30
**Clear and consise study with promising prelimirary results**

**Rating:** 3
**Confidence:** 2

**Review:**

Image-to-image translation in the context of super-resolution and domain-adaptation is performed by a 3D-RevGAN model. The authors add reversible blocks to the 3D-RevGAN model, without increasing memory cost. This property is demonstrated in the experiments, along with slightly improved results in the context of chest CT super-resolution according to lower MAE and higher PSNR (SSIM slightly lower) and in the context of chest CT domain adaptation with all scores impoved.

Pros:
- The manuscript is clearly and concisely written with balanced details in all sections.
- Experiment design seems adequate and the 3D-RevGAN results are promising.

Cons:
- The super-resolution example is rather basic, since the network is simply trained to “reverse engineer” a homogeneous down-sampling kernel, which seems to be a trivial task. Similar statement can be made for the domain-adaptation task. A pertinent example would be to perform domain adaptation across different scanner, possibly from different vendors.
- The number of image used for training in both tested application contexts is rather small (17/18). How does it compare to the number of free parameters? Albeit these preliminary results are promising, it remains to be shown how generalizable the results are on a larger datasets.

---

### Official Review · AnonReviewer1 · 2019-04-30
**Strong submission of novel model for memory efficient training with 3D data**

**Rating:** 4
**Confidence:** 2

**Review:**

The authors present a novel model using 3D Reversible GANs applied to super-resolution and domain-adaptation of CT chest data. The summarized methods reference the authors' full paper on arXiv, while the  results presented in the abstract are for different datasets to those presented in their full paper.

The authors show qualitative results for super-resolution using their Reversible GAN with a paired loss (from Pix2pix) as well as unpaired loss (from CycleGAN). Quantitative results are also shown for the super-resolution as well as domain-adaptation tasks, demonstrating better results with the paired loss and additional reversible blocks. While baseline models are not compared against in the abstract, the full paper results show that RevGAN performs better than the baseline CycleGAN or Pix2pix models for 3D volume super-resolution in most experiments.

The authors demonstrate that using their reversible GAN, even with an increasing network depth, the GPU memory requirements remain relatively constant. This is beneficial for training on volumetric medical image data, and allows deeper networks, larger batch sizes and larger image patches to be used, which could benefit many volumetric image analysis tasks.

While the methods and GPU memory savings are of interest, the qualitative and quantitative results are limited in their importance, although the authors do make clear that this work is a proof of principle that their method works.

The abstract is clearly written, and the novel methods as well as presented memory gains for applications using 3D medical images make this work of great interest to the medical imaging community.

---

### Decision · Program_Chairs · 2019-05-06
**Acceptance Decision**

Accept